# Structural basis for effector protein recognition by the Dot/Icm Type IVB coupling protein complex

Hyunmin Kim [1], Tomoko Kubori[2], Kohei Yamazaki [2,5], Mi-Jeong Kwak [1,6], Suk-Youl Park[3], Hiroki Nagai[2], Joseph P. Vogel [4] & Byung-Ha Oh [1✉]

The *Legionella pneumophila* Dot/Icm type IVB secretion system (T4BSS) is extremely versatile, translocating ~300 effector proteins into host cells. This specialized secretion system employs the Dot/Icm type IVB coupling protein (T4CP) complex, which includes IcmS, IcmW and LvgA, that are known to selectively assist the export of a subclass of effectors. Herein, the crystal structure of a four-subunit T4CP subcomplex bound to the effector protein VpdB reveals an interaction between LvgA and a linear motif in the C-terminus of VpdB. The same binding interface of LvgA also interacts with the C-terminal region of three additional effectors, SidH, SetA and PieA. Mutational analyses identified a FxxxLxxxK binding motif that is shared by VpdB and SidH, but not by SetA and PieA, showing that LvgA recognizes more than one type of binding motif. Together, this work provides a structural basis for how the Dot/Icm T4CP complex recognizes effectors, and highlights the multiple substrate-binding specificities of its adaptor subunit.

[1] Department of Biological Sciences, KAIST Institute for the Biocentury, Korea Advanced Institute of Science and Technology, Daejeon 34141, Republic of Korea. [2] Department of Microbiology, Graduate School of Medicine, Gifu University, 1-1 Yanagido, Gifu 501-1194, Japan. [3] Pohang Accelerator Laboratory, POSTECH, Pohang, Gyeongbuk 37673, Republic of Korea. [4] Department of Molecular Microbiology, Washington University School of Medicine, St. Louis, MO 63110, USA. [5] Present address: Veterinary Public Health, Kitasato University, Higashi 23-35-1, Towada, Aomori 034-8628, Japan. [6] Present address: CKD Research Institute, Yongin, Gyeonggi 16995, Republic of Korea. ✉email: bhoh@kaist.ac.kr

Prokaryotic pathogens subvert normal physiology of eukaryotic hosts by secreting effector proteins to achieve a successful infection. *Legionella pneumophila*, a Gram-negative intracellular pathogen causing Legionnaires' disease, secretes about 300 effector proteins into human macrophage cells through specialized protein-conducting channels. These proteins suppress immune defense and affect cellular homeostasis toward the intracellular survival, growth, and replication of the bacterium[1–3]. This type of protein secretion system involves three steps: (i) recognition of effector proteins by a coupling protein complex, (ii) unfolding of the effector proteins in the cytoplasm, and (iii) translocation of the effector proteins in an unfolded state through a transenvelope conduit[4–7]. *L. pneumophila* has a Dot/Icm type IVB secretion system (T4BSS)[7–9], which is composed of 26 Dot/Icm (defective for organelle trafficking/intracellular multiplication defect) proteins and the adaptor protein LvgA[10–14]. It is made up of two major complexes, the core transenvelope conduit and a Dot/Icm type IV coupling protein (T4CP) complex. The core transenvelope conduit, composed of ~10 Dot/Icm proteins, serves as the channel for effector translocation[15,16]. In the Dot/Icm T4CP complex, DotL/IcmO is the central subunit and belongs to the TraG family of coupling proteins that are classified as AAA+ type ATPases and function to link translocating substrates to the secretion conduit[4,17,18]. Unlike the prototypical T4CP TraG, DotL contains a ~200 residue C-terminal extension. While a similar region exists in many T4CPs, they vary greatly in length and amino acid sequence[19]. The C-terminal extension (CTE) of DotL interacts with a heterodimer of IcmS and IcmW (called IcmSW), DotN (IcmJ), and LvgA[4,19]. In addition, DotL interacts with the inner membrane protein DotM (IcmP) through its transmembrane helices and its membrane-proximal ATPase domain[4,20]. Thus, this multi-protein T4CP complex, rather than the DotL T4CP alone, would function to recognize and process effector proteins[21]. We previously reported that a pentameric complex, composed of a fragment of the DotL CTE, DotN, IcmS, IcmW, and LvgA, interacted with a number of *Legionella* effector proteins, showing that this C-terminal assembly of DotL is the substrate-recognition module of the Dot/Icm T4CP complex[19]. Together with crystallographic and modeling studies, we put forth a pseudo-atomic model for the Dot/Icm T4CP holocomplex, which is a hexamer of an N-terminal ATPase domain of DotL and a substrate-recognition assembly built on the CTE of DotL[19].

To examine how the Dot/Icm T4CP complex can recognize so many effector proteins, we extended our structural studies to examine the interaction between the Dot/Icm T4CP subcomplex and substrates. Herein, we report the crystal structure of a DotL(656-783)-IcmSW–LvgA complex bound to VpdB(461-590), a C-terminal fragment of the *Legionella* effector protein VpdB, which is sufficient for translocation in vivo. The structure provides the view of how the Dot/Icm T4CP complex recognizes its substrate protein and a glimpse at the molecular mechanisms underlying the recognition of a large number of effector proteins by this complex.

## Results

### Structure of a T4CP subcomplex bound to a C-terminal region of VpdB. VpdB is composed of an N-terminal phospholipase domain (residues 1–343) and a C-terminal domain (residues 344–598) with unknown function according to a sequence analysis by HHpred[22]. We generated five truncated constructs of VpdB: VpdB(11-590), VpdB(11-488), VpdB(11-343), VpdB(425-590), and VpdB(461-590) (Fig. 1a). Except VpdB(11-343), all the other four constructs exhibited clearly detectable interaction with DotL(656-783)–IcmSW–LvgA in a native polyacrylamide gel electrophoresis (PAGE)-based binding assay (Fig. 1a), indicating

that the binding interface for the T4CP subcomplex resides on the C-terminal domain of VpdB, but not on the enzymatic domain. Subsequently, complexes between DotL(656-783)–IcmSW–LvgA and each of the four constructs of VpdB were produced and subjected to crystallization. Of these, DotL(656-783)–IcmSW–LvgA bound to VpdB(461-590) was crystallized, and the structure of this complex was determined at 2.8 Å (Supplementary Table 1). VpdB(461-590) is composed of five α-helices, and four of them (α2–α5; residues 489–590) form a four-helical bundle (Fig. 1b). The first helix (α1; residues 461–488) is almost completely separated from the four-helix bundle and interacts with a wide and shallow groove, contoured by α3, α4 and nearby loops of the LvgA subunit. This intermolecular interaction is fairly extensive, involving 14 VpdB residues: Gln466, Leu469, Lys470, Lys472, Thr473, Met474, Phe476, Lys477, Arg479, Leu480, Gln481, Phe483, Lys484, and Glu487 on α1 within 4.5 Å from LvgA, and burying 1700 Å² surface area of LvgA. Of these, six residues are involved in considerable interactions: Lys470, Thr473, Phe476, Leu480, Phe483, and Lys484 (Fig. 1c). The asymmetric unit of the crystal contains two copies of the complex, and they exhibit virtually the same intermolecular interactions. No subunit other than LvgA in DotL(656-783)–IcmSW–LvgA interacts with VpdB(461-590), highlighting a critical role of LvgA in the recognition of VpdB. This crystallographic observation is consistent with a previous observation that DotL(656-783)–IcmSW–LvgA binds to VpdB, but DotL(656-783)–IcmSW does not[19].

The four-helix bundle is involved in minor contacts with LvgA in comparison with that between α1 and LvgA (Fig. 1c). Indeed, in a native PAGE-based protein-binding assay, α1 of VpdB alone interacted with DotL(656-783)–IcmSW–LvgA, while the four-helical bundle of VpdB did not (Fig. 1d), confirming that α1 of VpdB is necessary and sufficient for LvgA binding.

### Key intermolecular interaction between LvgA and VpdB. Along the binding interface of α1, Phe476 of VpdB appeared to make a prominent interaction with a hydrophobic pocket formed by four hydrophobic residues (Phe149, Ile153, Pro166, Tyr173) of LvgA (Fig. 2a). To test whether this observed hydrophobic interaction contributes significantly to the intermolecular interaction, we generated a near full-length VpdB variant containing a F476E mutation, VpdB(11-590;F476E), and a DotL(656-783)–IcmSW–LvgA(I153E) complex, which contained an I153E mutation on the hydrophobic pocket of LvgA. In a native PAGE-based protein binding assay, VpdB(11-590;F476E) did not exhibit a detectable interaction with the wild-type DotL(656-783)–IcmSW–LvgA complex. Likewise, the mutant DotL(656-783)–IcmSW–LvgA (I153E) complex neither exhibited a notable interaction with the wild-type version of VpdB(11-590) (Fig. 2b). These results were consistent with a quantitative protein binding assay employing bio-layer interferometry; neither the DotL(656-783)–IcmSW–LvgA and VpdB(11-590;F476E) pair nor the DotL(656-783)–IcmSW–LvgA(I153E) and VpdB(11-590) pair exhibited a detectable binding signal, while the DotL(656-783)–IcmSW–LvgA and VpdB(11-590) pair interacted with each other with an apparent dissociation constant ($K_D$) of around 400 nM (Fig. 2c). Thus, the hydrophobic interaction between Phe476 of VpdB and the hydrophobic pocket of LvgA plays a significant role in the intermolecular recognition between the two proteins.

Next, to identify the key interacting residues of VpdB, we generated six VpdB(11-590) variants containing an alanine substitution of a set of residues (Lys470, Thr473, Phe476, Leu480, Phe483, Lys484) that are located on α1 and involved in notable interactions with LvgA (Fig. 1c). The intermolecular interactions between these variants and DotL

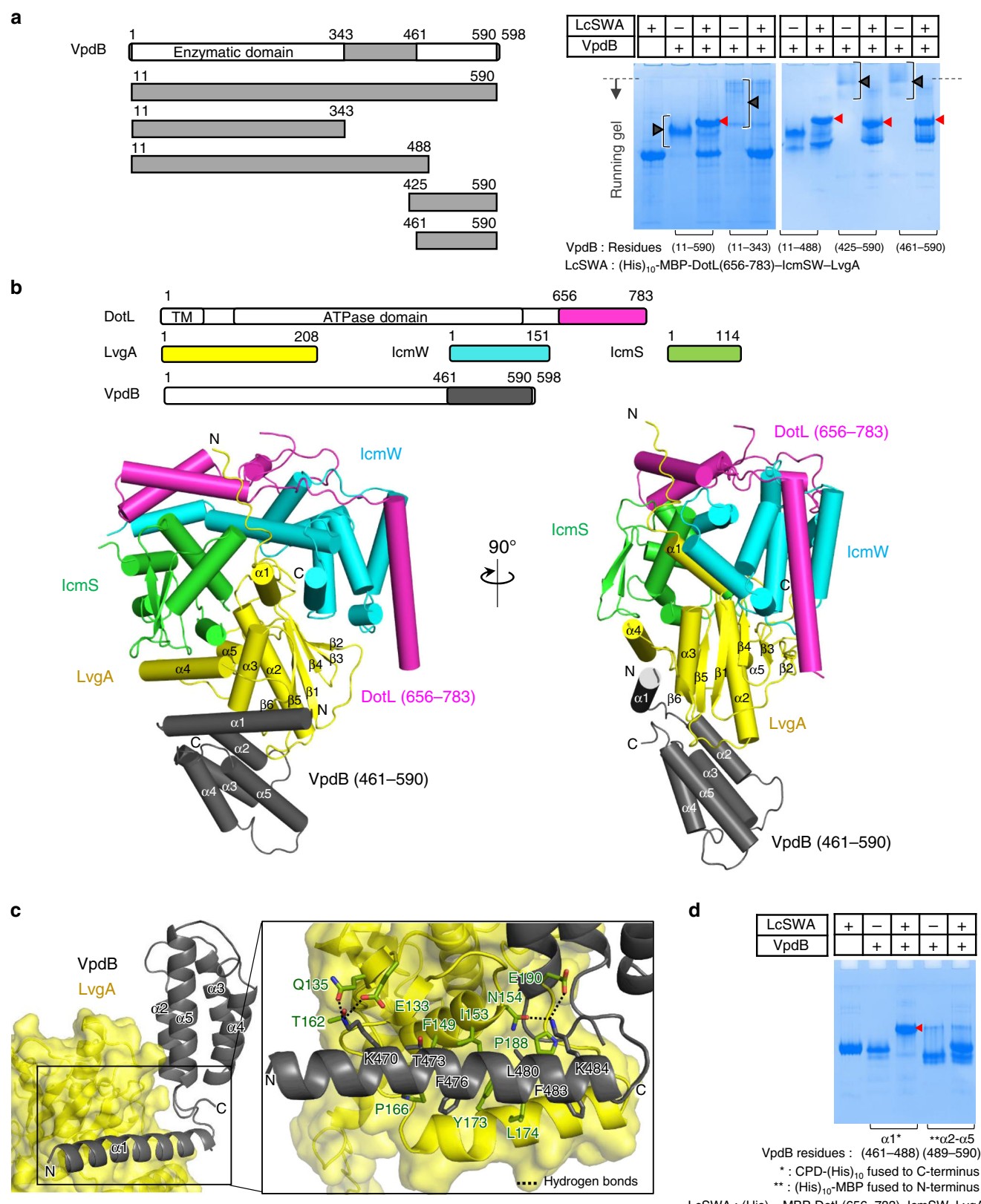

(656-783)–IcmSW–LvgA were quantified by bio-layer interferometry (Fig. 2d). The F476A mutation resulted in the most drastic decrease in the binding affinity ($K_D > 15\,\mu M$). The L480A and K484A mutations were the second and third most affecting mutations, increasing $K_D$ value by 14 and 9 times, respectively.

The other three substitutions affected the binding affinity, but to a lesser extent (Fig. 2d). Together, these mutational analyses highlight the importance of the hydrophobic pocket of LvgA, and identify the three most important residues on α1 in the recognition of VpdB by the adaptor subunit.

**Fig. 1 Crystal structure of DotL(656-783)–IcmSW-LvgA–VpdB(461-590). a** VpdB(461-590) binds to (His)$_{10}$-MBP-DotL(656-783)–IcmSW–LvgA. Each of the five truncated VpdB proteins (10 μM) was incubated with (His)$_{10}$-MBP-DotL(656-783)–IcmSW–LvgA at a 1:1 molar ratio and visualized on a native PAGE-gel by Coomassie staining. Schematic drawings of the constructs of VpdB are shown on the left. The black brackets and the red arrowheads indicate the diffusive input protein bands and newly formed protein bands, respectively. The dotted line indicates the borderline between the stacking and the running gel. **b** Two views of structure of the DotL(656-783)–IcmSW–LvgA–VpdB(461-590). Schematic drawings of the five protein constructs are shown at the top. Protein constructs used for structure determination are color coded and labeled. **c** Enlarged view of the interaction between VpdB(461-590) and LvgA. α1 of VpdB(461-590) lies on a wide and shallow groove of LvgA. VpdB residues majorly involved in the intermolecular interaction are shown as sticks. **d** Helix α1 is the major binding motif. (His)$_{10}$-MBP-DotL(656-783)–IcmSW–LvgA (10 μM) was incubated with CPD-(His)$_{10}$ tagged helix α1 or (His)$_{10}$-MBP tagged four-helix bundle α2–α5 of VpdB(461-590) at a 1:1 molar ratio, and visualized on a native gel. Only helix α1 exhibited a newly formed protein band indicated by the red arrowhead. All the native PAGE analyses were repeated more than 3 times.

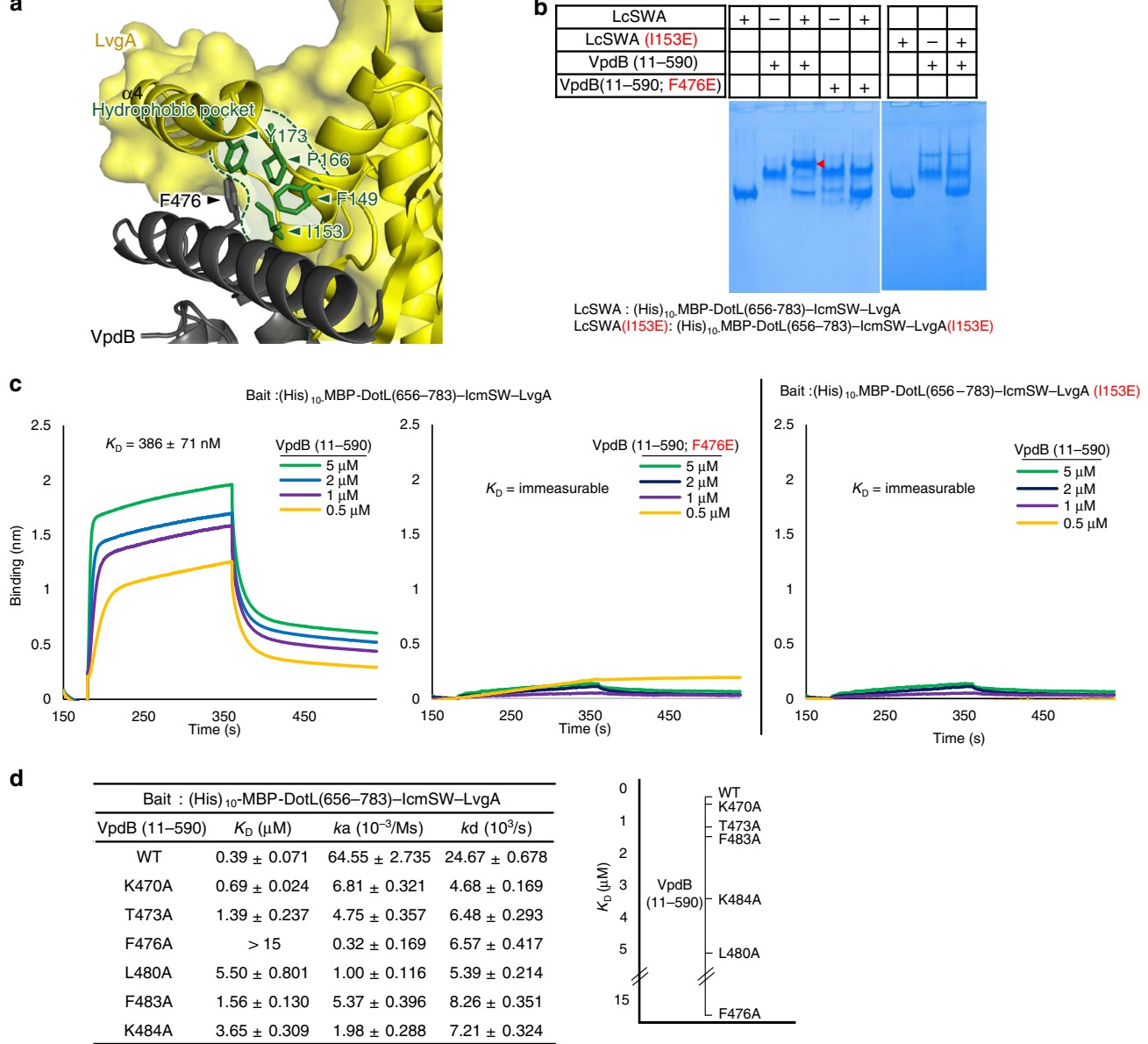

**Fig. 2 Interaction between α1 of VpdB(461-590) and the LvgA subunit. a** A prominent hydrophobic interaction. Phe476 of VpdB packs against a hydrophobic pocket on LvgA indicated by the dotted line. Four residues that constitute the hydrophobic pocket are shown as sticks. **b** Native PAGE analysis. The VpdB(F476E) or LvgA(I153E) mutation disrupted the interaction between VpdB(11-590) and (His)$_{10}$-MBP-DotL(656-783)–IcmSW–LvgA. **c** Quantification. The binding affinity between (His)$_{10}$-MBP-DotL(656-783)–IcmSW–LvgA and VpdB(11-590) was drastically decreased by either VpdB (F476E) or LvgA(I153E) mutation. The interaction was measured by bio-layer interferometry, where the indicated bait was immobilized on a Ni-NTA biosensor via the N-terminal (His)$_{10}$-MBP tag fused to DotL(656-783). The measurement was triplicated. **d** Alanine mutagenesis. Binding affinities between (His)$_{10}$-MBP-DotL(656-783)–IcmSW–LvgA and VpdB(11-590) variants containing the indicated alanine substitution were analyzed by bio-layer interferometry. A plot of the measured $K_D$ values is shown on the right. Quantification was duplicated for each variant.

**Recognition of the effectors SetA, PieA, and SidH**. Previously, we showed that two other Dot/Icm substrates, SetA and PieA interact with DotL(656-783)–IcmSW–LvgA, but not with DotL (656-783)–IcmSW[19]. To locate the binding interface of these two effectors, the N-terminal enzymatic domain and the C-terminal domain of the proteins were produced and subjected to the native PAGE-based protein-binding assay. The C-terminal domains of these proteins, SetA(480-644) and PieA(513-699), exhibited clearly detectable interactions with DotL(656-783)–IcmSW–LvgA, similarly as full-length SetA and PieA. In contrast, the N-terminal enzymatic domains of SetA(1-273) and PieA(1-349) displayed no notable interaction (Fig. 3a). Likewise, a C-terminal fragment of the effector SidH, SidH(1830-2200), also exhibited clearly detectable interaction with DotL(656-783)–IcmSW–LvgA (Fig. 3a), as observed previously[19]. Thus, like VpdB, all of the three effector proteins interact with DotL(656-783)–IcmSW–LvgA via a C-terminal region.

Next, we examined whether the hydrophobic pocket of LvgA is also involved in the intermolecular recognition. On a native gel, SetA, PieA, and SidH(1830-2200) interact significantly weaker with the DotL(656-783)–IcmSW–LvgA(I153E) variant in comparison with their interactions with the wild-type subcomplex (Fig. 3b). Consistently, quantification of these interactions by biolayer interferometry showed that the I153E mutation on LvgA resulted in immeasurable binding affinity for SetA and about 13-fold or 11-fold decreased binding affinity for PieA and SidH, respectively, in terms of $K_D$ (Fig. 3c). Likely, the crystallographically elucidated VpdB-binding groove on LvgA, including the hydrophobic pocket, is the common and main interface for the recognition of SetA, PieA, and SidH.

**FxxxLxxxK binding motif in VpdB and SidH**. SidH exhibits no significant overall sequence similarity with VpdB(461-590). However, a nine-amino acid stretch (residues 2191–2199 of SidH) possesses high sequence identity to a portion of helix α1 of VpdB (461-590). In this segment of SidH, Phe2191, Leu2195, and Lys2199 correspond to Phe476, Leu480, and Lys484 of VpdB, respectively, which are the three most important residues for the interaction with LvgA. (Fig. 2d). We generated three SidH(1830-2200) variants containing an alanine substitution of these three residues. All of the three variants exhibited decreased interaction with DotL(656-783)–IcmSW–LvgA in a native PAGE analysis (Fig. 3d). Consistently, SidH(1830-2200;F2191A), SidH(1830-2200;L2195A), and SidH(1830-2200;K2199A) displayed at least 25-, 24-, or 4-fold increased $K_D$ value, respectively, in comparison with the wild-type version of SidH(1830-2200) (Fig. 3d). To know whether SidH might interact with the LvgA subunit via an α-helix containing Phe2191, Leu2195, and Lys2199, a SidH(2183-2200) construct was generated, which is predicted to be a continuous single α-helix and encompasses the nine-residue stretch that aligns with helix α1 of VpdB(461-590). This 18-residue α-helix interacted with DotL(656-783)–IcmSW–LvgA in a native PAGE-based protein binding assay (Fig. 3e). Together with the parallel results observed for helix α1 of VpdB (Fig. 2), these data indicate that the LvgA subunit recognizes a FxxxLxxxK motif on an α-helix. Unlike SidH, the C-terminal domains of SetA and PieA that bind to the T4CP complex do not contain a complete FxxxLxxxK motif, although each possesses a FxxxI/V sequence motif on a predicted α-helix, which could correspond to $F^{476}xxxL^{480}$ in VpdB and $F^{2191}xxxL^{2195}$ in SidH (Supplementary Fig. 1a). Since the phenylalanine residue is the most important binding residue in VpdB, we introduced a Phe-to-Ala mutation in the $^{619}FxxxV^{623}$ motif in SetA and the $^{618}FxxxI^{622}$ motif in PieA. However, neither mutation affected the binding interaction with DotL(656-783)–IcmSW–LvgA (Supplementary Fig. 1b),

suggesting that SetA and PieA interact with the LvgA subunit differently, even though the interaction is dependent on the same hydrophobic pocket of LvgA (Fig. 3b, c). Narrowing down the LvgA-binding sequences of SetA and PieA by deletion mutagenesis was hampered by the instability of the constructs shorter than SetA(480-644) or PieA(513-699).

**Secretion of VpdB and SidH depend on IcmSW and partially on LvgA**. VpdB is one of the *Legionella* effector proteins whose secretion into host cells has been uncharacterized. We performed CyaA translocation assay[23] where the reporter *Bordetella pertussis* adenylate cyclase (CyaA) fused to VpdB was used, and the secretion of the fusion protein into the host Chinese hamster ovary cells overexpressing FcγRII (CHO FcγRII) was monitored by measuring cAMP production. We employed five strains: Lp01, a derivative of *L. pneumophila* (Philadelphia-1); Lp01 Δ*dotA*, an isogenic mutant lacking *dotA* and defective in the Dot/Icm transporter function; Lp01 Δ*icmSW*, a mutant lacking *icmS* and *icmW*; Lp01 Δ*lvgA*, a mutant lacking *lvgA*; Lp01 Δ*icmSW*Δ*lvgA*, a mutant lacking *icmS, icmW* and *lvgA*. Like RalF, the translocation of VpdB from Lp01 Δ*dotA* was drastically affected, demonstrating that VpdB is a substrate of the Dot/Icm T4BSS (Fig. 4a, d). However, unlike RalF, whose translocation is independent of IcmSW[24,25], translocation of VpdB from Lp01 Δ*icmSW* or Lp01 Δ*lvgA* was attenuated significantly, in comparison with that from the wild-type strain Lp01 (Fig. 4a, d). The absence of IcmS and IcmW (in Lp01 Δ*icmSW*) resulted in about 30-fold decrease in the cAMP level, while that of LvgA (in Lp01 Δ*lvgA*) resulted in about 10-fold decrease (Fig. 4a). The data demonstrate that VpdB is an IcmSW-dependent effector for its translocation, and that the VpdB translocation depends also on LvgA, but less heavily than it does on IcmSW. The C-terminal VpdB(461-590) fragment was sufficient for the translocation, since it exhibited nearly the same translocation pattern as full-length VpdB. In contrast, the N-terminal VpdB(1-343) fragment did not exhibit a demonstrable translocation (Fig. 4a). To know whether the intermolecular interaction between Phe476 and the hydrophobic pocket of LvgA observed in the crystal structure is important for VpdB translocation (Fig. 2b, c), the translocation assay was performed with VpdB(F476E) fused to CyaA. Notably, the translocation of this mutant from Lp01 (Fig. 4a, 4th set) was similarly attenuated as that of wild-type VpdB from Lp01 Δ*lvgA* (Fig. 4a, 1st set), supporting that the key intermolecular interaction observed in vitro is directly relevant for the translocation of VpdB in vivo.

SidH and its variants exhibited a translocation pattern quite similar to VpdB. Both full-length SidH and a C-terminal fragment SidH(1830-2200) depended on IcmSW and LvgA for translocation; but the effect of deletion of *lvgA* was less significant than that of *icmSW* (Fig. 4b), thus suggesting that VpdB and SidH share the same translocation mechanism.

Of note, translocation of VpdB and SidH from Lp01 Δ*icmSW* is similar to that from Lp01 Δ*icmSW*Δ*lvgA*, indicating that the additional deletion of *lvgA* has no appreciable effect, and thus their LvgA-dependent translocation is subject to the presence of IcmSW (Fig. 4c). These data are consistent with the crystal structure of DotL(656-783)–IcmSW–LvgA–VpdB(461-590) (Fig. 1b), showing that the presence of IcmSW is required for LvgA to assemble into the T4CP complex and function in this complex.

**LvgA-dependent secretion of PieA and SetA**. Additionally, we performed the translocation assay for SetA and PieA, the latter of which is known to depend on IcmSW for its translocation[26]. In this assay, translocation of SetA was also attenuated by the

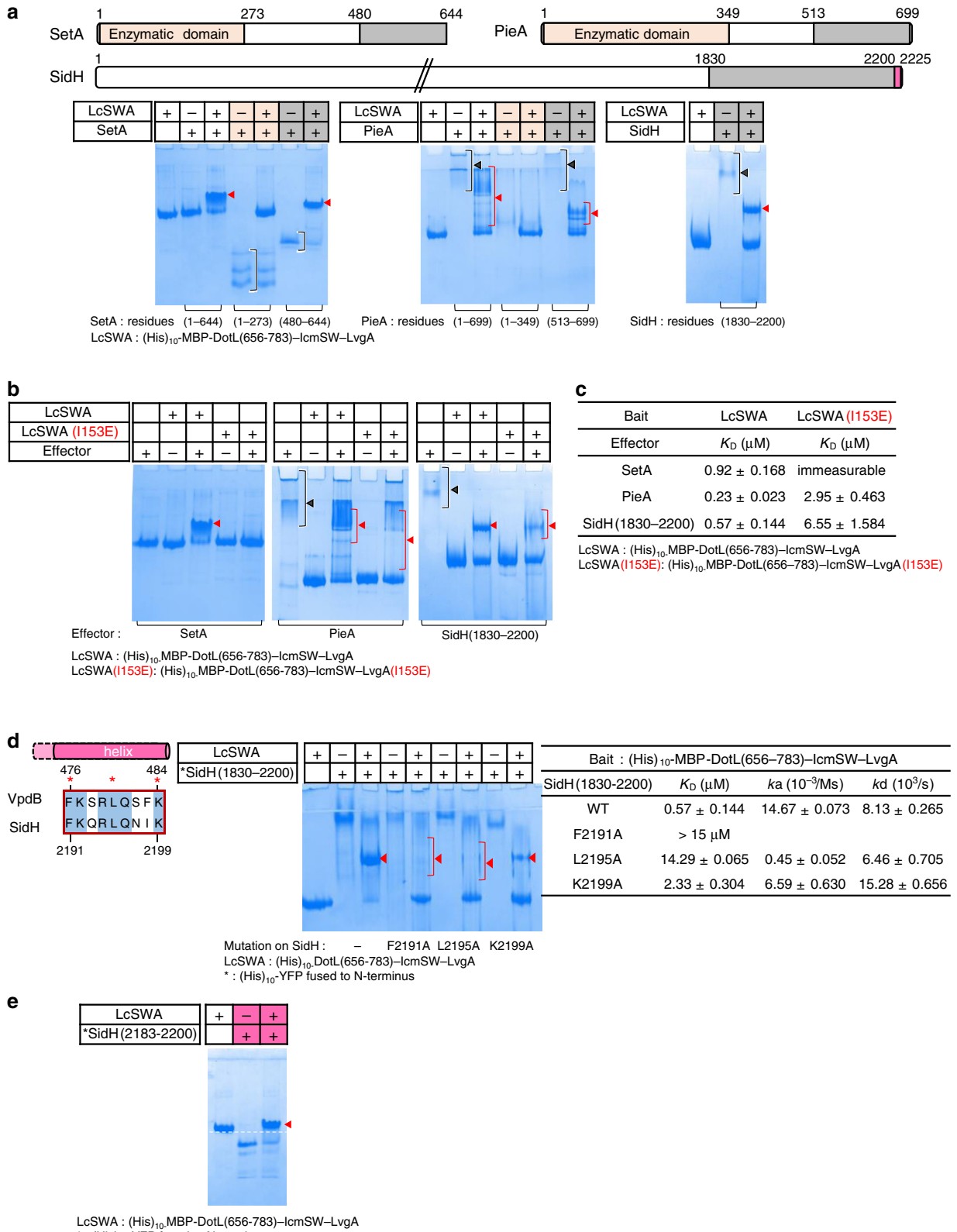

absence of *icmSW* or *lvgA*. The pattern of the attenuation was similar to that of VpdB: a greater attenuation displayed by Lp01 Δ*icmSW* than by Lp01 Δ*lvgA* (Fig. 4d). In comparison, the translocation of PieA appeared to depend specifically on LvgA, because the translocation from Lp01 Δ*icmSW* was as attenuated as that from Lp01 Δ*lvgA* (Fig. 4d).

**A model for the Dot/Icm T4CP holocomplex bound to full-length VpdB**. In the structure of VpdB(461-590) bound to DotL (656-783)–IcmSW–LvgA, helix α1 is largely separated from the four-helical bundle, which is likely to result from a binding-induced large conformational change. Conceivably, the C-terminal domain of VpdB is conformationally flexible in free

**Fig. 3 SetA, PieA, and SidH interaction with DotL(656-783)–IcmSW–LvgA. a** Native PAGE analysis. Not the enzymatic domain but the C-terminal domain of SetA, PieA, and SidH interacted with (His)$_{10}$-MBP-DotL(656-783)–IcmSW–LvgA, as indicated by the newly formed protein bands (red arrowheads). Schematic drawings of the three effectors are shown at the top. Black brackets indicate diffusive input protein bands. **b** Importance of the hydrophobic pocket on LvgA. Native PAGE analyses were performed with the indicated complex containing an I153E mutation. Its interactions with SetA, PieA, and SidH were disrupted or significantly reduced. **c** Quantification by bio-layer interferometry. The I153E substitution on the hydrophobic pocket of LvgA decreases the binding affinity drastically for SetA and significantly for PieA and SidH (~13-fold and ~11-fold increase in $K_D$, respectively). Each quantification was repeated 2 or more times. **d** Identification of the binding motif in SidH. A C-terminal segment of SidH (residues 2183–2200) bears sequence similarity with α1 helix of VpdB, and is predicted to be an α-helix (left). The three indicated mutations in SidH(1830-2200) reduced the binding affinity for DotL(656-783)–IcmSW–LvgA in a varying degree (middle and right). The measurement was repeated 2 or more times for each variant. **e** FxxxLxxxK motif containing α-helix of SidH as a major binding fragment. (His)$_{10}$-YFP-SidH(2183-2200) (10 μM) was incubated with (His)$_{10}$-MBP-DotL (656-783)–IcmSW–LvgA at a 1:1 molar ratio, and visualized on a native gel. The red arrowhead indicates the formation of a new protein band. All the native PAGE analyses were repeated more than 3 times.

VpdB. To gain insight into the substrate recognition by the Dot/Icm T4CP holocomplex, we constructed a model for the holocomplex bound to full-length VpdB. The structure of DotL(656-783)–IcmSW–LvgA–VpdB(461-590) was superposed onto a hexameric model of the Dot/Icm T4CP holocomplex which was built based on a homology model of the ATPase domain of DotL and the crystal structures of the DotL CTE in complex with other subunits[19]. Next, we used the Robetta server[27,28] to build a model for full-length VpdB in the LvgA-bound state. As templates for homology modeling, we employed the structures of VpdB(461-590) and the N-terminal domain of VipD (residues 1–412), which is a *Legionella* effector protein that shares high sequence homology with the N-terminal domain of VpdB[29–31] (Fig. 5). As expected, the junction between the N- and C-terminal domains of VpdB was heterogeneously modeled, and their relative orientation varied greatly among the five output homology models (Supplementary Fig. 3). One of them exhibited no steric clash upon structural superposition onto the Dot/Icm T4CP holocomplex model, and showed that the C-terminal four-helix bundle of VpdB points toward the central chamber-like space of the holocomplex whereas the N-terminal domain of VpdB faces the outside of the holocomplex (Fig. 5). The constructed model shows that the bound substrate is far from the membrane-proximal ATPase assembly, and suggests that a hinge-bending motion of the substrate-recognition assembly is required for the bound substrate to reach the ATPase assembly for processing (Fig. 5).

## Discussion

LvgA, a virulence factor important for intracellular growth of *Legionella*[32,33], is a newly identified subunit for the Dot/Icm T4CP complex[19]. This subunit interacts with at least four effector proteins (VpdB, SidH, SetA, and PieA). We define LvgA as an adaptor, because it is required for linking these effector proteins to the rest of the complex. Of note, DotL(656-783)–IcmSW–LvgA exhibited no detectable binding interaction with other effectors such as SidJ or Lpg0393[19], indicating that a subset of *Legionella* effectors are recognized by the LvgA subunit. To serve as an adaptor, LvgA has a hydrophobic surface that binds to a concave surface of IcmSW[19], whereas a shallow, wide surface containing a hydrophobic pocket on the opposite side serves as the interface for binding effector proteins (Figs. 1b, 2a–c, and 3b).

To date, two different translocation signals in *Legionella* effectors have been identified. One is the C-terminal extreme sequence containing a leucine or other hydrophobic residue, as observed for RalF whose secretion is independent of IcmSW[25]. The other is the E-block motif (EExxE) near the C-terminus, which was demonstrated with SidM[34]. The E-block motif was proposed to be recognized by the Dot/Icm T4CP complex via the DotM subunit, which contains basic patches that may serve as the binding platform for the E-block[20].

In this study, we identified the FxxxLxxxK motif on a C-terminal α-helix of VpdB and SidH that is recognized by the adaptor subunit LvgA in the Dot/Icm T4CP complex (Figs. 1c, d, 2d, and 3d, e). 257 out of a total of 2930 proteins of the *L. pneumophila* strain Philadelphia-1 (Proteome ID: UP000000609) contain the FxxxLxxxK sequence motif. Of these, 46 proteins, including SidH and VpdB, belong to the 280 known *Legionella* effectors[35], indicating that this sequence is enriched in the effector proteins (16.4% of effectors vs. 8.0% of non-effectors). However, the FxxxLxxxK motif is not present in the C-terminal regions of SetA and PieA, which can also bind to LvgA. The C-terminal regions of SetA and PieA, which are composed mostly of α-helices, do not share sequence homology with each other (Supplementary Fig. 1a). Therefore, they likely present an alternative motif(s) that interacts with the binding interface of LvgA.

Since VpdB(461-590) interacts only with LvgA in the crystal structure of DotL(656-783)–IcmSW–LvgA–VpdB(461-590), VpdB translocation from the Lp01 Δ*lvgA* strain was expected to be as attenuated as that from the Lp01 Δ*icmSW* strain. However, export was more diminished in the Lp01 Δ*icmSW* strain compared to the Lp01 Δ*lvgA* strain (Fig. 4a), indicating that VpdB translocation is partially dependent on LvgA. We hypothesize that additional, as-yet-unidentified, adaptor proteins may exist that compensate in the absence of LvgA. In contrast, strains lacking IcmSW would not display any adaptor, thus explaining the greater reduction of VpdB translocation observed in the Lp01 Δ*icmSW* mutant. The same scenario would be true for SetA translocation, but not for PieA translocation. LvgA appears to be the sole adaptor for PieA, because its translocation from Lp01 Δ*icmSW* was as attenuated as that from Lp01 Δ*lvgA* (Fig. 4b).

We further postulate that the Dot/Icm T4CP complex is likely to be heterogeneous in terms of the adaptor displayed, where the six copies of IcmSW bound to the DotL hexamer are either unoccupied, occupied by LvgA or by an unidentified adaptor protein(s). This heterogeneity in adaptors would mediate an expanded specificity of the Dot/Icm T4CP complex and allow recognition of a wide range of substrate proteins. Unoccupied IcmSW would recruit effector proteins that interact directly with IcmSW, such as SdeA, which contains a 200-residue domain near its C-terminus capable of binding to DotL(656-798)-IcmSW[36]. On the other hand, IcmSW bound to LvgA or other adaptors would recruit effector proteins that are able to interact with their cognate adaptor(s). In this scenario, IcmSW is of greater importance than LvgA or other adaptor protein(s) in effector translocation, which explains why the *Legionella* strains lacking *icmS* or *icmW* are more defective than the strain lacking *lvgA* in the intracellular growth of *Legionella*[33]. The adaptor proteins would have multiple specificity of binding, as observed for LvgA, in order to recognize not a single effector but a subset of effector proteins. Based on the large number of effectors exported by the Dot/Icm system, we suspect that *Legionella* may encode an

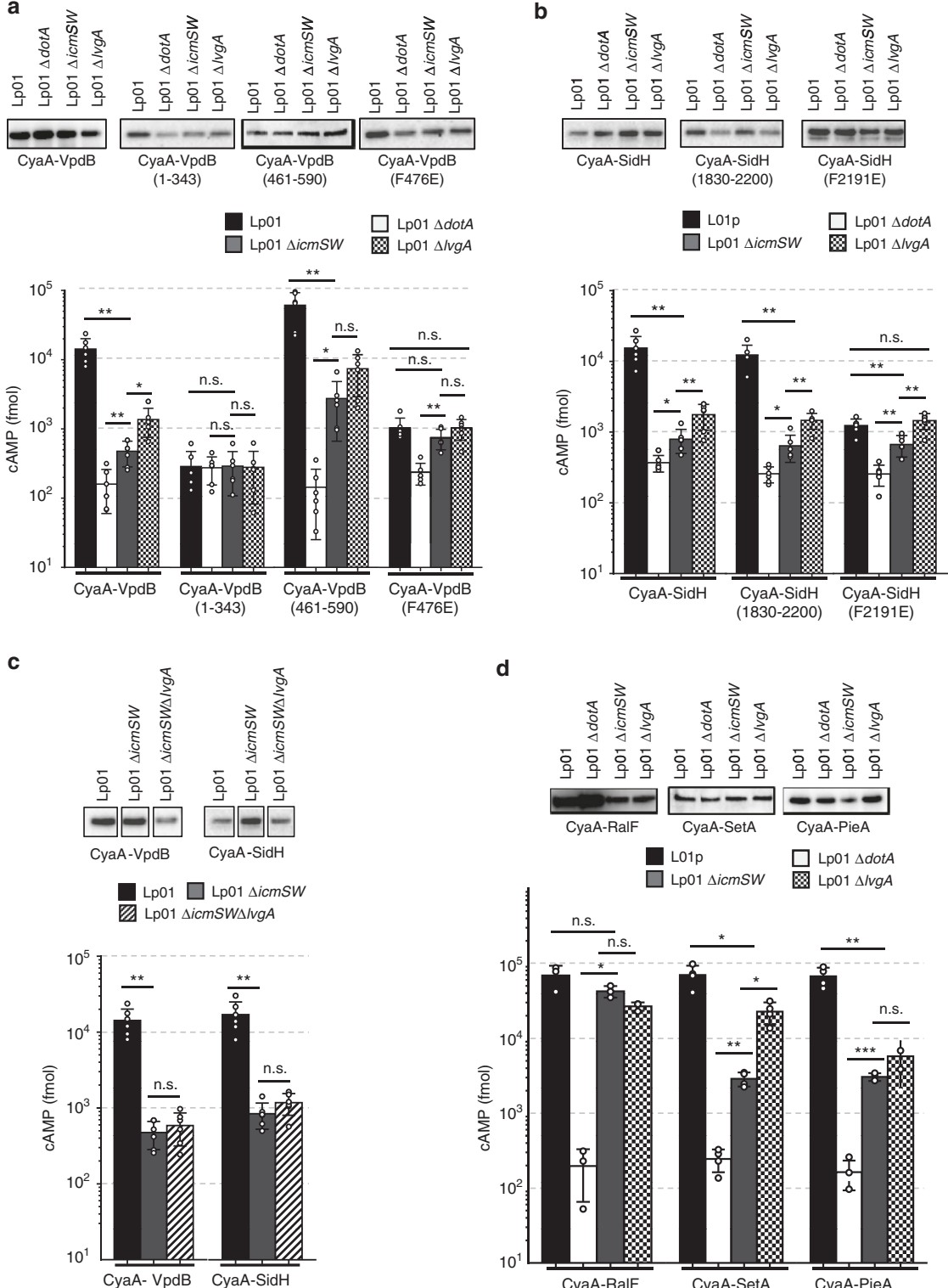

arsenal of adaptors similar to LvgA that are essential to the regulation of export of T4BSS substrates by this pathogen.

In conclusion, the presented work provides the atomic-resolution view of how the Dot/Icm T4CP complex recognizes effector proteins to be translocated. The adaptor subunit LvgA utilizes a wide and shallow binding surface with a hydrophobic pocket to recognize a subset of cognate effectors. This surface exhibits multiple substrate-binding specificity, as it binds as-yet-unknown motif(s) as well as the FxxxLxxxK motif on an α-helix, both located near the C-terminus of the four effector proteins tested in this study. We suggest that the Dot/Icm T4CP complex is equipped with a number of different mechanisms to recognize ~300 effector proteins, including compositional heterogeneity involving more adaptor proteins other than LvgA.

**Fig. 4 CyaA translocation assay. a** Translocation of VpdB. CHO-FcγRII cells were infected with the *Legionella* strains expressing the indicated VpdB constructs fused to CyaA, and the cAMP production was measured. Shown at the top is protein immunoblotting of CyaA-VpdB expressed in each *Legionella* strain using anti-CyaA IgG. The expression level of the tested VpdB constructs was similar. Each experiment was biologically duplicated and technically triplicated. **b** Translocation of SidH. Translocation patterns of SidH and its variants are similar to those of VpdB. Translocation assay and immunoblotting were performed similarly as in (**a**). **c** Dependency of LvgA-mediated translocation on IcmSW. Lp01 Δ*icmSW* and Lp01 Δ*icmSW*Δ*lvgA* strains exhibited a similar level of attenuation in the transport of VpdB and SidH. **d** LvgA-dependent translocation of SetA and PieA. Both effectors depended on LvgA to be translocated. CyaA-RalF was used as a control. Each experiment was triplicated, including biological duplication. For immunoblotting, 10 milliOD$_{600}$ units were loaded. Statistical differences compared with the results obtained with Lp01 Δ*icmSW* were determined by the two-sided *t*-test for the other strains. The error bars represent mean values ± SD. \*\*\**p* < 0.001; \*\**p* < 0.01; \**p* < 0.05; n.s., not significant. Exact *p*-values are provided in Supplementary Fig. 4.

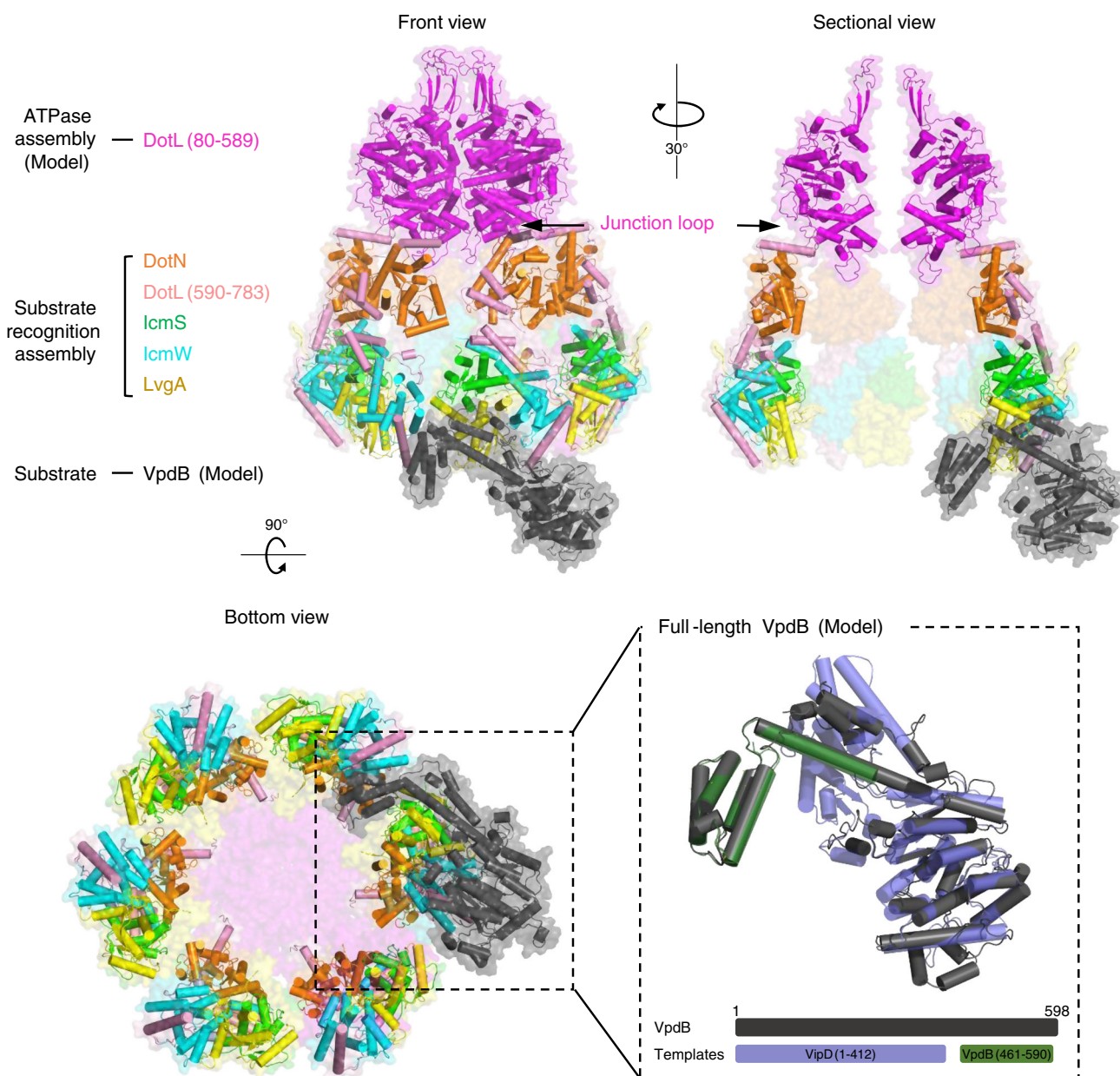

**Fig. 5 A model for substrate binding to Dot/Icm T4CP holocomplex.** Three different views of the Dot/Icm T4CP holocomplex bound to full-length VpdB are shown. The junction loop between the ATPase assembly and the substrate recognition assembly is indicated, where a hinge bending motion might take place. A model for full-length VpdB built by homology modeling is shown. Each subunit is color coded. The holocomplex model does not include the membrane segment of DotL and the DotM subunit. The *Legionella pneumophila* Dot/Icm type IVB secretion system (T4BSS) translocates effector proteins into host cells and the recognition of these effectors is mediated by the Dot/Icm type IV coupling protein (T4CP) complex. Here, the authors present the crystal structure of a four-subunit containing T4CP subcomplex bound to the effector protein VpdB and identify a FxxxLxxxK binding motif that is present in a subset of the effectors and which is recognized by the T4CP adaptor subunit LvgA.

## Methods

**Protein production and purification**. A DNA fragment coding for LvgA (AAU26621.1) was cloned into a modified version of the pET 22b vector (Novagen) (Amp$^R$), and those for IcmS (AAU26539.1) and IcmW (AAU28746.1) were cloned into the pET 22b vector (Kan$^R$). The two vectors were introduced into the *Escherichia coli* BL21 (DE3) RIPL strain (Novagen), and the proteins were expressed at 18 °C overnight after induction by 200 μM isopropyl β-D-1-thioga-lactopyranoside (IPTG).

A DNA fragment coding for residues 656–783 of DotL (AAU26543.1) was cloned into pET 22b vector, and expressed with a C-terminally fused cysteine protease domain (CPD) of the *Vibrio cholerae* MARTX toxin with a (His)$_{10}$ tag from *E. coli* BL21 (DE3) RIPL strain. Cells were grown overnight at 18 °C after 200 μM IPTG induction. Cell lysates in Buffer A (20 mM Tris–HCl (pH 7.5), 100 mM NaCl) were co-sonicated, cleared, and applied onto HisPur Cobalt Resin (Thermo Scientific). The CPD-(His)$_{10}$ tag was cleaved on-gel by inducing autolytic cleavage with 100 μM sodium phytate (Sigma Aldrich). The flow-through fraction contained DotL(656-783)–IcmSW–LvgA, and was further purified using a Hitrap Q anion exchange column (GE Healthcare) and employing a linear 50–1000 mM NaCl gradient in 20 mM Tris (pH 7.5).

A DNA fragment encoding residues 461–590 of VpdB (WP_010946959) was cloned into a modified version of the pET 22b vector (Novagen), and the protein was expressed as a fusion protein with an N-terminal (His)$_{10}$-MBP tag in the *E. coli* BL21 (DE3) RIPL strain (Novagen) at 18 °C overnight after 200 μM IPTG induction. Cell lysate was applied onto HisPur Cobalt Resin (Thermo Scientific), washed with Buffer A (20 mM Tris–HCl (pH 7.5), 100 mM NaCl), and eluted with Buffer A supplemented with additional 200 mM imidazole. The eluate was treated with the tobacco etch virus protease overnight, and was further purified using a Hitrap Q anion exchange column (GE Healthcare) and employing a linear 50–1000 mM NaCl gradient in 20 mM Tris (pH 7.5). The purified VpdB(461-590) was mixed with DotL(656-783)–IcmSW–LvgA at 1:1 molar ratio and the mixture was loaded onto a HiLoad 26/60 Superdex 75 gel-filtration column (GE Healthcare) to isolate the DotL(656-783)–IcmSW–LvgA–VpdB(461-590) complex with Buffer A supplemented with 50 mM β-mercaptoethanol.

All the truncated VpdB constructs except for VpdB(461-488) were expressed as a fusion protein containing an N-terminal (His)$_{10}$-MBP tag. Truncated variants of SidH (AAU28877) were expressed with an N-terminally fused (His)$_{10}$-YFP tag. All the described constructs of SetA (Q5ZU30), PieA (YP_095979.1), and VpdB(461-488) were expressed with a C-terminally fused CPD-(His)$_{10}$ tag. The procedures for expression and purification of these proteins were similar to those used for DotL(656-783)–IcmSW–LvgA and VpdB(461-590). A buffer solution containing 20 mM Tris–HCl (pH 7.5) and 300 mM NaCl was used to purify the PieA constructs instead of buffer A. VpdB(461-488)-CPD-(His)$_{10}$, (His)$_{10}$-MBP-VpdB(489-590), and (His)$_{10}$-YFP-SidH(2183-2200) were used for protein binding assay without cleaving the tag. All the primers used are listed in Supplementary Table 2.

**Crystallization, structure determination, and refinement**. DotL(656-783)–IcmSW–LvgA–VpdB(461-590) (25 mg/ml) was crystallized in a solution containing 50 mM sodium cacodylate (pH 7.0) and 1.8 M ammonium sulfate, at 20 °C. The crystals were soaked in the crystallization solution supplemented with 15% glycerol for cryoprotection. X-ray diffraction data were processed with the *HKL2000* suit[37]. The structure of DotL(656-783)–IcmSW–LvgA (PDB code: 5X90 [https://www.rcsb.org/structure/5X90]) was used as a search model for the phase determination with the program *PHASER*[38]. *COOT*[39] and *CNS*[40] were used to perform model building and structure refinement (Supplementary Table 1).

**Bio-layer interferometry**. The $K_D$ values for protein–protein interaction were determined by using a BLItz instrument (ForteBio). Either (His)$_{10}$-MBP-DotL(656-783)–IcmSW–LvgA or (His)$_{10}$-MBP-DotL(656-783)–IcmSW–LvgA(I153E) in buffer A was loaded onto a Ni-NTA biosensor tip for 120 s. The biosensor tips were incubated in 500 μl of buffer A for 30 s, loaded with 4 μl of VpdB, SidH, or SetA for 180 s (association step), and again incubated in the same buffer for 180 s (dissociation step). To measure the binding affinity between PieA and (His)$_{10}$-MBP-DotL(656-783)–IcmSW–LvgA or (His)$_{10}$-MBP-DotL(656-783)–IcmSW–LvgA (I153E), a buffer solution containing 20 mM Tris–HCl (pH 7.5) and 300 mM NaCl was used. BLItz Pro software (ForteBio) was used to analyze the binding kinetics.

**CyaA translocation assay**. CHO FcγRII cells were replated in 24-well tissue culture plates one day prior to infection. The cells were incubated with the culture medium (αMEM + 10% fetal bovine serum) containing additional rabbit anti-*Legionella* antiserum (1:3000 ratio) for 30 min before infection. *Legionella* strains were added to each well ($3.0 \times 10^6$ bacteria per well). The bacterial cells were precipitated onto a confluent monolayer of the host cells ($1 \times 10^5$ cells per well) by centrifugation for 5 min at 200×g. Plates were immediately warmed in a 37 °C water bath for 5 min, then placed in a CO$_2$ incubator at 37 °C for 1 h. Cells were washed 3 times with ice-cold phosphate-buffered saline and lysed in 250 μl of 100 mM HCl on ice. The cAMP level was determined for each extract by using an EIA system (Amersham Biosciences, RPN-2255).

**Protein immunoblotting assay**. Whole cell lysates of each *Legionella* strain expressing a CyaA-tagged effector were used. Anti-CyaA mouse monoclonal IgG (sc-13582, Santa Cruz Biotechnology, Inc.) diluted to 1:3000 and goat anti-mouse IgG (#31430, Invitrogen) conjugated to horseradish peroxidase diluted to 1:10,000 were used as a primary and secondary antibody, respectively. TOPview™ ECL Pico Plus (Enzynomics, Inc.) was used for detection.

**Homology modeling**. The modeling was performed on the Robetta server[28], using the N-terminal portion of the VipD structure (PDB 4AKF; residues 1–412 [https://www.rcsb.org/structure/4akf]) and the structure of VpdB(461-590) as templates. Five models were obtained as the output structures.

**Reporting summary**. Further information on research design is available in the Nature Research Reporting Summary linked to this article.

## Data availability

The coordinates of the structure of DotL(656-783)–IcmSW–LvgA–VpdB(461-590) are deposited in the Protein Data Bank (PDB code: 7BWK). The source data underlying Figs. 1, 2b–d, 3, and 4 and Supplementary Figs. 1 and 2 are provided as a Source Data file. Other data are available from the corresponding author upon reasonable request.

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

## Acknowledgements

The X-ray data were collected at the beamline NE-3A at the Photon Factory, Japan and at the beamline 11C at the Pohang Accelerator Laboratory, Korea. This work was supported by the National Research Foundation of Korea (NRF-2018R1A2B3004764).

## Author contributions

Experimental design: B.-H.O., H.K., H.N., and J.P.V. Data acquisition: H.K., T.K., K.Y., and S.-Y.P. Data analysis: H.K., M.-J.K., T.K., and H.N. Manuscript writing: B.-H.O. and H.K.; Manuscript editing: B.-H.O., H.K., H.N., and J.P.V. Supervision: B.-H.O.

## Competing interests

The authors declare no competing interests.
