## [Peer Review File · Nature Communications]

Reviewers' comments:

Reviewer #1 (Remarks to the Author):

In this manuscript B-H Oh follow their structural characterization of the coupling complex of Legionella Dot/Icm sT4BSS secretion system with the visualization of how an effector protein is recognized and binds to this complex. They determined the structure of the DotI(656-783)-IcmS-IcmW-LvgA with the VpdB C-terminal segment VpdB(461-590). The binding is facilitated by the interaction of an α -helix from VpdB with a hydrophobic groove in the exposed LvgA surface. The N-terminal, catalytic domain of VpdB does not participate in the recognition by the secretion apparatus. The structure of the complex suggested several residues as significantly contributing to the binding. Further mutational analysis led to identification of three residues essential for binding and established the FxxxLxxxK sequence as a binding motif recognized by LvgA. This motif exists in 12 other effectors and for one of them, SidH, the binding was experimentally confirmed. Importantly, this motif is absent in two other effectors that were previously shown to bind to LvgA and it suggests that LvgA recognizes the effectors in more than one way. The hypothesis that there is another adaptor protein that might also recognize VpdB although binds it less efficiently nicely explains the translocation data with Legionella mutants with delete either LvgA or IcmSW. This is a very substantial contribution to our understanding of the T4SS secretion machinery and provides an insight into how Legionella has adapted to the task of recognizing a large number of effectors to be secreted by the same machinery.

There are several places where more explanations are necessary. The native PAGE gels provide the essential data for interaction of T4CP complex with various effector constructs yet they are not adequately described. For example, in Fig 1a why are there still lower band in lanes containing LcSWA and VpdB? If they were mixed in a 1:1 ratio then either only the complex is present or we have in addition also free VpdB and LcSWA, the latter in equimolar amounts. Also for example, for the (425-590) construct the VpdB lane is empty but there are two bands in the next lane containing this construct and LcSWA. What are these bands? Similarly, in Fig 1d the a17 aa fragment of VpdB runs at the same distance as LcSWA (lanes 1 and 2) and the mixture shows a very large shift, much larger, it appears, than addition of a larger VpdB fragment in Fig 1a. p. 6, l. 107 and on: The residues on VpdB that interact with LvgA are explicitly indicated but the LvgA residues involved in these interactions are not mentioned. There are several charged and hydrophilic residues on VpdB so there are not only hydrophobic interaction involved. The more detailed view of the interacting residues should be provided. A view of the Lys484 is one of the key sidechain electrostatic potential of both interacting surfaces would also be helpful. Lys484 is a key component of the binding motif so it is important to describe why this is so and what it interacts with.

p. 9, l. 170: "exhibited no detectable or significantly decrease interaction..." - "interact significantly weaker ..."

Reviewer #2 (Remarks to the Author):

The manuscript by Kim et al continues the analysis of the DotLc-IcmS-IcmW-LvgA complex and its interaction with Legionella pneumophila effectors. The first part of the manuscript focuses on the effector protein VpdB, which includes the structure of the above mentioned complex together with the C-terminal part of VpdB, nicely showing the direct interaction between a specific alpha helix of VpdB and LvgA. In the second part, three additional effectors (SetA, PieA and SidH) were added to the analysis showing that they also require LvgA for interaction with the complex, and the same amino acid of LvgA (I152E) required for interaction with VpdB is also required for the interaction with these three effectors. In addition, a "consensus" sequence present only in VpdB and SidH was identified. In the third part, the authors tried to confirm their results using the CyaA translocation assay using the same effectors and deletion mutants in the different complex components as well

as a mutation in VpdB. The third part instead of confirming or validating the results of the first two parts raises several concerns, as listed below.

Major comments

1. The difference between the effect of an *icmSW* deletion mutant (100-fold) and the one of the *lvgA* deletion mutant (10-fold) on the translocation of VpdB, does not fit the observation that LvgA makes the direct contact with VpdB. The authors described a compelling theory about a number of adaptor proteins which might function like LvgA and connect different groups of effectors to the IcmSW complex. This is an interesting theory, but there is no proof in the manuscript that it is correct. Due to the results mentioned above, the author should at the least show that all the effect on translocation mediated by LvgA indeed occurs via the IcmSW complex. For this, the translocation of VpdB (and the other effectors) should be examined from an *icmSW-lvgA* triple deletion mutant that should result in no additive effect on translocation in comparison to the *icmSW* deletion mutant.

2. Due to the same problem described above, SidH should also be included in the translocation analysis. The homology between VpdB and SidH (Fig. 3d) predicts that both effectors will show a similar result as far as the effect of *icmSW*, *lvgA* and the triple deletion mutant and this will strengthen the manuscript. This is also required since the other two effectors examined (*SetA* and *PeiA*) do not have the "consensus" described.

3. The CyaA translocation analysis included the analysis of VpdB-F476E which showed lower degree of translocation in comparison to the wild-type VpdB (Fig. 4a). The translocation of this mutant was similar to the one of the wild-type VpdB when examined in the *lvgA* mutant. The authors should examine this mutant for translocation from the *lvgA* mutant to confirm that *lvgA* have no effect on the translocation of this mutant, since if an additive effect will be observed, it will indicate that additional factors are involved.

4. The translocation of the CyaA vector control should be included, or it should be indicate that the degree of translocation obtained with the CyaA vector when examined in the wild-type strain is similar to the one obtained from the *dotA* deletion mutant containing the effectors.

5. The effectors examined utilize the IcmSW-LvgA for translocation but they probably also use the C-terminal secretion signal. This is evident from the results presented in Fig. 4 showing that the degrees of translocation from the *dotA* mutant were much lower in comparison to the ones obtained with *icmSW* deletion. The authors should consider the option that deleting the C-terminal secretion signal from one of these effectors and examining this truncated effector in the wild-type strain and the *icmSW*, *lvgA* and *icmSW-lvgA* mutants will give a more clearer result since only the system under study will be involved in the translocation of the effector and not the C-terminal secretion signal.

Reviewer #3 (Remarks to the Author):

The Dot/Icm complex is a major virulence factor of *Legionella*, yet the mechanism by which the 300 effectors are recognized, selected for and loaded into the translocon is poorly understood. The authors provide novel and valuable insight into the molecular basis for the interaction between two effectors and the Dot/Icm coupling protein complex (T4CP). The authors provide compelling biochemical, biological and structural evidence that a consensus binding motif (FxxxLxxxK) in an alpha helix at the effector C-terminus interacts with T4CP adaptor protein LvgA. They identified specific residues in both the effector and adaptor proteins that are predominantly involved in mediating these interactions, including a hydrophobic pocket in LvgA that accommodates the phenylalanine residue in the effector binding motif. The effector binding motif FxxxLxxxK is present in 10 other effectors suggesting common mechanisms of recognition and loading. In

contrast, the effectors SetA and PieA, for which translocation is also LvgA-dependent, lack the consensus sequence suggesting the existence of a second LvgA-dependent interacting motif. This manuscript was very well written, a pleasure to read.

Major concerns:

1. The authors state that "more than 300 Legionella proteins contain the FxxxLxxxK motif. Of these, 12 effector proteins...contain this sequence motif on a predicted α -helix located near their C-terminus" (lines 305-308). Please clarify this statement – does this mean that none of the other >288 proteins have this motif in an alpha-helix? Are the authors suggesting that despite this motif being enriched in non-effector proteins ($288/3000 = 10\%$ vs $12/300 = 4\%$), this motif only functions as a targeting signal in effectors, and this is determined by its location/position? Based on Figure S2, the motif is not restricted to the protein C-terminus (MavK, LegAU13, Lpg1484), the location within the alpha-helix varies (N- vs C-terminal portion) and at least in the case of MavK, overlaps with a B-sheet. Thus there are a number of additional variables to consider when defining a bona fide signal - positioning along the length of the protein, positioning within alpha-helices versus B-sheets, positioning within the alpha-helix itself (VpdB and SidH both have the motif in the C-terminal end of a C-terminally located helix). The data would be more informative in defining this interaction and class of effectors if these parameters were more thoroughly discussed and/or investigated, as the mere presence of this motif is not predictive of a targeting signal (which the authors allude to).

2. Given the high degree of variability for some protein products in the native gels - multiple bands (SetA, Fig. 3a), laddering effects (PieA, Fig. 3a,b) and difference in size and/or relative abundance (SidH, Fig. 3d) - the interaction data in some cases is a little difficult to interpret. The data would be more convincing if the stability of the full length proteins were demonstrated by denaturing PAGE. Moreover, SidH K2199A shows only a 4-fold increase in KD over WT but the variant protein migrates differently than the WT protein which could indicate structural differences. If so, the diminished interaction could be due to indirect effects on protein folding rather than amino acid substitutions in the binding motif.

3. The authors propose that the difference in VpdB translocation efficiency between *icmSW* and *lvgA* mutants is the presence of a second VpdB chaperone. However, an alternative explanation is that VpdB translocation is separately *IcmSW*- and *LvgA*-dependent. Is translocation of VpdBF476E the same or decreased in the *icmSW* mutant compared to the *lvgA* mutant?

4. Please indicate the number of biological and technical replicates of experiments and provide statistical analyses.

Minor comments

1. Please check labels in Fig. 2b for the second gel, it appears that VpdB and LcSWA(I153E) "+" may be switched.

2. Using consistent amino acid nomenclature (single letter rather than mixing single and 3 letter abbreviations) would be helpful.

3. "3000 folds" and "10000 folds" line 420-421 should read 1:3000 and 1:10,000 respectively.

Point-by-Point Responses

We are very grateful for the constructive comments from the three reviewers. In revising the manuscript, we marked the changes with red letters.

Reviewer #1:

1. There are several places where more explanations are necessary. The native PAGE gels provide the essential data for interaction of T4CP complex with various effector constructs yet they are not adequately described.

1-a. For example, in Fig 1a why are there still lower band in lanes containing LcSWA and VpdB? If they were mixed in a 1:1 ratio then either only the complex is present or we have in addition also free VpdB and LcSWA, the latter in equimolar amounts.

→

The indicated lane contains three proteins: the LcSWA–VpdB(11-590) complex, free LcSWA and free VpdB(11-590). LcSWA migrates as a narrow band, whereas VpdB(11-590) migrates as a diffusive band. Therefore, the band intensity of free LcSWA (the lower band) appears higher than that of VpdB(11-590). However, the integrated intensity of the diffusive band would be similar to that of the narrow band. We now label the diffusive VpdB(11-590) band with the 'j' mark in Fig. 1a.

1-b. Also for example, for the (425-590) construct the VpdB lane is empty but there are two bands in the next lane containing this construct and LcSWA. What are these bands?

→The lanes are not empty. Please note that the protein bands of VpdB(425-590) and VpdB(461-590) are smeared widely and thus faint. However, these two constructs both exhibit a narrow protein band upon formation of a complex with LcSWA, which is likely to result from disorder-to-order change of helix α_1 , as observed in the crystal structure. We now label these diffusive bands with the 'j' mark in Fig. 1a.

Similarly, in Fig 1d the a17 aa fragment of VpdB runs at the same distance as LcSWA (lanes 1 and 2) and the mixture shows a very large shift, much larger, it appears, than addition of a larger VpdB fragment in Fig 1a.

→

These two short VpdB fragments were tagged with a cysteinyl protease domain-(His)₁₀ or a (His)₁₀-maltose binding protein for stabilization. Thus, the size of these constructs are much bigger than the untagged helix fragments. While the tagging was stated in the maintext, it was not in the figure. We now indicate the presence of the tag in Fig. 1d. Nevertheless, the band positions of the complexes cannot be explained by the additional size of the tag, but can be explained by the fact that the mobility of a protein or protein complex on a native gel depends not only on the size but also on the net charge of the analyte.

Additional response to the Comment #1; The shape, intensity and position of a protein band on a native gel often are not clearly interpretable, unlike those on a denaturing polyacrylamide gel. In parallel with the native PAGE-based protein-protein interaction analysis, we carried out bio-layer interferometry. The results of this protein-protein interaction analysis are consistent with our interpretation of the native PAGE gels.

2. p. 6, l. 107 and on: --, The residues on VpdB that interact with LvgA are explicitly indicated but the LvgA residues involved in these interactions are not mentioned. There

are several charged and hydrophilic residues on VpdB so there are not only hydrophobic interaction involved. The more detailed view of the interacting residues should be provided. A view of the Lys484 is one of the key sidechain electrostatic potential of both interacting surfaces would also be helpful. Lys484 is a key component of the binding motif so it is important to describe why this is so and what it interacts with.

→

We now revised Fig. 1c to address this concern. The new figure shows the LvgA residues interacting with the key interface residues of VpdB, including Lys484.

3. p. 9, l. 170: “exhibited no detectable or significantly decrease interaction...” - “interact significantly weaker ...”

→

Corrected.

Reviewer #2:

1. The difference between the effect of an *icmSW* deletion mutant (100-fold) and the one of the *lvgA* deletion mutant (10-fold) on the translocation of VpdB, does not fit the observation that LvgA makes the direct contact with VpdB. The authors described a compelling theory about a number of adaptor proteins which might function like LvgA and connect different groups of effectors to the IcmSW complex. This is an interesting theory, but there is no proof in the manuscript that it is correct. Due to the results mentioned above, the author should at the least show that all the effect on translocation mediated by LvgA indeed occurs via the IcmSW complex. For this, the translocation of VpdB (and the other effectors) should be examined from an *icmSW-lvgA* triple deletion mutant that should result in no additive effect on translocation in comparison to the *icmSW* deletion mutant.

→

We generated a triple knockout strain, Lp01 $\Delta icmSW\Delta lvgA$ and performed translocation assays for VpdB and SidH. The results show no additive effect by the additional deletion of *lvgA*; translocation of SidH and VpdB were similarly attenuated in both Lp01 $\Delta icmSW$ and Lp01 $\Delta icmSW\Delta lvgA$ compared to the wild-type Lp01. Therefore, LvgA-dependent effector translocation is subject to the presence of IcmSW, as suggested by the crystal structure of LcSWA–VpdB(461-590). Now the results are shown in Fig. 4c and stated in the main text.

2. Due to the same problem described above, SidH should also be included in the translocation analysis. The homology between VpdB and SidH (Fig. 3d) predicts that both effectors will show a similar result as far as the effect of IcmSW, *lvgA* and the triple deletion mutant and this will strengthen the manuscript. This is also required since the other two effectors examined (SetA and PeiA) do not have the “consensus” described.

→

As per reviewer’s comment, we performed the translocation assay for full-length SidH, a C-terminal fragment SidH(1830-2200) and the SidH(F2191E) mutant. Translocation of full-length SidH and SidH(1830-2200) exhibited a similar pattern: attenuation in Lp01 $\Delta lvgA$ and more attenuation in Lp01 $\Delta icmSW$ compared to the wild-type Lp01, as was similarly observed for VpdB. Translocation of the SidH(F2191E) mutant from the wild-type Lp01 was attenuated to a level comparable to that from Lp01 $\Delta lvgA$, as was similarly observed for VpdB and the VpdB(F476E) mutant (Fig. 4a). Thus, LvgA and SidH appear to share the same translocation mechanism. Now the results are shown in Fig. 4b and stated in the main text.

3. The CyaA translocation analysis included the analysis of VpdB-F476E which showed lower degree of translocation in comparison to the wild-type VpdB (Fig. 4a). The translocation of this mutant was similar to the one of the wild-type VpdB when examined in the *lvgA* mutant. The authors should examine this mutant for translocation from the *lvgA* mutant to confirm that *lvgA* have no effect on the translocation of this mutant, since if an additive effect will be observed, it will indicate that additional factors are involved.

→

As per reviewer's comment, we performed the translocation assay for VpdB(F476E) with Lp01 and Lp01 Δ *lvgA*, showing no appreciable additional effect of the *lvgA* deletion (Fig. 4a).

4. The translocation of the CyaA vector control should be included, or it should be indicate that the degree of translocation obtained with the CyaA vector when examined in the wild-type strain is similar to the one obtained from the *dotA* deletion mutant containing the effectors.

→

This has been already established and published (PNAS January 18, 2005 102 (3) 826-831). A relevant figure is shown below. Translocation of CyaA alone from Lp01 is similar to that of CyaA-RalF from Lp01 Δ *dotA* strain (B, lane 3 and C, lane 8). Since we used the same strain and virtually the same vector, this control experiment was not performed.

5. The effectors examined utilize the IcmSW-LvgA for translocation but they probably also use the C-terminal secretion signal. This is evident from the results presented in Fig. 4 showing that the degrees of translocation from the *dotA* mutant were much lower in comparison to the ones obtained with *icmSW* deletion. The authors should consider the option that deleting the C-terminal secretion signal from one of these effectors and examining this truncated effector in the wild-type strain and the *icmSW*, *lvgA* and *icmSW-lvgA* mutants will give a more clearer result since only the system under study will be involved in the translocation of the effector and not the C-terminal secretion signal.

→

To address this comment, we generated a N-terminal fragment of VpdB, VpdB(1-343), and analyzed its translocation from each strain. A notable translocation of this construct was not observed in all four strains (WT, Δ *dotA*, Δ *icmSW*, Δ *lvgA*), supporting our

conclusion that the identified sequence motif at the C-terminal domain is involved in the translocation of the effector. The results are shown in Fig. 4a.

Reviewer #3:

Major

1. The authors state that “more than 300 *Legionella* proteins contain the FxxxLxxxK motif. Of these, 12 effector proteins....contain this sequence motif on a predicted α -helix located near their C-terminus” (lines 305-308). Please clarify this statement – does this mean that none of the other >288 proteins have this motif in an alpha-helix? Are the authors suggesting that despite this motif being enriched in non-effector proteins ($288/3000 = 10\%$ vs $12/300 = 4\%$), this motif only functions as a targeting signal in effectors, and this is determined by its location/position? Based on Figure S2, the motif is not restricted to the protein C-terminus (MavK, LegAU13, Lpg1484), the location within the alpha-helix varies (N- vs C-terminal portion) and at least in the case of MavK, overlaps with a B-sheet. Thus there are a number of additional variables to consider when defining a bone fide signal - positioning along the length of the protein, positioning within alpha-helices versus B-sheets, positioning within the alpha-helix itself (VpdB and SidH both have the motif in the C-terminal end of a C-terminally located helix). The data would be more informative in defining this interaction and class of effectors if these parameters were more thoroughly discussed and/or investigated, as the mere presence of this motif is not predictive of a targeting signal (which the authors allude to).

→

We agree with the reviewer that the predicative statement based on the identification of this sequence motif in the 12 effectors is premature and misleading. In the revised manuscript, we deleted the sentences, “More than 300 *Legionella pneumophila* proteins contain the FxxxLxxxK motif. Of these, 12 effector proteins, including SidH and VpdB, contain this sequence motif on a predicted α -helix located near their C-terminus (Supplementary Fig. 3)” and Supplementary Fig. 3.

2. Given the high degree of variability for some protein products in the native gels - multiple bands (SetA, Fig. 3a), laddering effects (PieA, Fig. 3a,b) and difference in size and/or relative abundance (SidH, Fig. 3d) - the interaction data in some cases is a little difficult to interpret. The data would be more convincing if the stability of the full length proteins were demonstrated by denaturing PAGE. Moreover, SidH K2199A shows only a 4-fold increase in KD over WT but the variant protein migrates differently than the WT protein which could indicate structural differences. If so, the diminished interaction could be due to indirect effects on protein folding rather than amino acid substitutions in the binding motif.

→

The band pattern on native gel is greatly affected by the isoelectric point, conformational homogeneity and etc. As Supplementary Fig. 2 we now present size-exclusion chromatography (to show the homogeneity of the proteins) and a SDS-PAGE run (to show the purity of the proteins) for the SetA and PieA constructs (in Fig. 3a) and the SidH(1830-2200) mutants (in Fig. 3d). While PieA(1-343) exhibited conformational heterogeneity (broad and multiple peaks), the other constructs were mostly homogenous. The denaturing gel shows that the purity of these proteins were high. We used sub-fractions containing right-size species for all biochemical analyses.

Full-length PieA and PieA(513-699) exhibited almost no mobility and smearing band patterns of the two proteins on the native gel (Fig. 3a). The narrow single elution peaks

of the two proteins (Supplementary Fig. 2) indicate that they were homogeneous. They have theoretical pI of 8.57 and 9.13, respectively, and these high pI values were likely responsible for their band patterns on the native gel. Likewise, in the case of the SidH(1830-2200) K2199A mutant, the notable mobility increase in comparison with the wild-type version is likely due to a PI change rather than to a structural difference, because the major peak from the chromatographic column exhibits the same elution volume between the two proteins. We believe that the inclusion of the SDS-PAGE gel and the elution profiles in the revised manuscript will help readers to interpret the native PAGE results.

3. The authors propose that the difference in VpdB translocation efficiency between *icmSW* and *lvgA* mutants is the presence of a second VpdB chaperone. However, an alternative explanation is that VpdB translocation is separately *IcmSW*- and *LvgA*-dependent. Is translocation of VpdBF476E the same or decreased in the *icmSW* mutant compared to the *lvgA* mutant?

→

This comment is related to the comments raised by Reviewer #2. The translocation of VpdB(F476E) (as well as the wild-type protein) from Lp01 $\Delta icmSW$ is not the same, but decreased compared to that from Lp01 $\Delta lvgA$ (Fig. 4a). The result, however, is not interpreted as an indication that there are two separate *IcmSW*- or *LvgA*-dependent translocation pathways. This is because the newly constructed Lp01 $\Delta icmSW\Delta lvgA$ strain exhibited no appreciable additional effect on the VpdB and SidH translocation compared to the Lp01 $\Delta icmSW$ double knockout strain (Fig. 4c). The data strengthen our conclusion that the *LvgA*-dependent effector translocation is subject to the presence of *IcmSW*.

4. Please indicate the number of biological and technical replicates of experiments and provide statistical analyses.

→

The bio-layer interferometry experiments were triplicated except for the VpdB variants containing an alanine substitution (duplicated). The *CyaA* translocation experiments were biologically duplicated and technically triplicated except for *RalF*, *PieA* and *SetA* (technically triplicated including biological duplication). The data were analyzed by (heteroscedastic) t-test, with the data from Lp01 $\Delta icmSW$ as a control group to show similarity between effector translocation from Lp01 $\Delta icmSW$ and that from Lp01 $\Delta icmSW\Delta lvgA$ or difference between translocation from Lp01 $\Delta icmSW$ and that from the other strains. Statistical labels are added on the graphs, and these details are now stated in the legends for Fig. 4.

Minor

1. Please check labels in Fig. 2b for the second gel, it appears that VpdB and LcSWA(I153E) "+" may be switched.

→

Pointing that out is appreciated. We now correct the labels.

2. Using consistent amino acid nomenclature (single letter rather than mixing single and 3 letter abbreviations) would be helpful.

→

We used the three-letter abbreviations to indicate amino acid residues, and one-letter

codes to indicate amino acid substitutions, which is a conventional way.

3. "3000 folds" and "10000 folds" line 420-421 should read 1:3000 and 1:10,000 respectively.

→

Corrected.

REVIEWERS' COMMENTS:

Reviewer #1 (Remarks to the Author):

The authors performed additional experiments to address the comments from the reviewers and clarified the text accordingly. This manuscript adds significantly to our understanding of the initial steps in the secretion through the T4SS. I have no further comments.

Reviewer #2 (Remarks to the Author):

The revised manuscript by Kim et al is an improved version of the previously submitted manuscript. Most of the major comments raised by this reviewer were answered thoroughly, beside one. In comment number five of the previous review an experiment was suggested in which the C-terminal secretion signal and the LvgA binding signal will be separated in a way that the C-terminal secretion signal will be deleted and the LvgA signal will be left intact. Such a truncated effector will give a clearer result in comparison to the constructs used, since only the secretion signal under study will be involved in the translocation of the effector and not the C-terminal secretion signal. In their response, the authors indicated that they used VpdB(1-343) to answer this comment. This construct does not answer the comment. As indicated by the authors in the discussion section, the C-terminal secretion signal was previously shown to be located at the very end of the effectors, as the last 25-30 amino acids. The VpdB(1-343) construct used is deleted for both the LvgA signal and the C-terminal signal and consequently this construct does not separate the two signals since it is deleted for both of them (and as expected this construct showed no translocation). The proper construct for the analysis suggested should have been VpdD(1-570) or VpdD(461-570), in which the C-terminal signal is deleted but the LvgA signal is present. Separation of the two signals will strengthen the manuscript, but it is not absolutely required for publication.

Reviewer #3 (Remarks to the Author):

In general, the authors have done a commendable job addressing the majority of the reviewer comments with additional experimentation and explanations.

Comments:

1. Simply removing the statement "more than 300 Legionella proteins contain the FxxxLxxxK motif. Of these, 12 effectors proteins....contain this sequence motif on a predicted α -helix located near their C-terminus" (line 316 in the revised manuscript) does not address the issue. It is important to note how enriched a translocation motif is in effectors compared to non-effector proteins as this speaks to specificity (and possibly a requirement for multiple signals?). The authors cite the E-block translocation motif for which enrichment in effectors was demonstrated. Here it seems to be the opposite ($288/3000 = 10\%$ of non-effectors vs $12/300 = 4\%$ of effectors).

Point-by-Point Responses_2

Reviewer #1: No further comments

Reviewer #2:

“----- The VpdB(1-343) construct used is deleted for both the LvgA signal and the C-terminal signal and consequently this construct does not separate the two signals since it is deleted for both of them (and as expected this construct showed no translocation). The proper construct for the analysis suggested should have been VpdD(1-570) or VpdD(461-570), in which the C-terminal signal is deleted but the LvgA signal is present. Separation of the two signals will strengthen the manuscript, but it is not absolutely required for publication.

→

We misunderstood the reviewer’s previous comment #5, and generated the VpdB(1-343) construct and carried out translocation experiment on it, which further supports the engagement of the newly identified C-terminal FxxxLxxxK sequence motif in the effector translocation. As the reviewer pointed out now, we needed to generate a VpdB(1-570) or a VpdB(461-570) construct to correctly address the previous comment. However, the reviewer also commented that it is not absolutely required for publication. Accordingly, we did not carry out further experiment, which otherwise will delay the publication really significantly.

Reviewer #3:

Simply removing the statement “more than 300 Legionella proteins contain the FxxxLxxxK motif. Of these, 12 effectors proteins....contain this sequence motif on a predicted α -helix located near their C-terminus” (line 316 in the revised manuscript) does not address the issue. It is important to note how enriched a translocation motif is in effectors compared to non-effector proteins as this speaks to specificity (and possibly a requirement for multiple signals?). The authors cite the E-block translocation motif for which enrichment in effectors was demonstrated. Here is seems to be the opposite ($288/3000 = 10\%$ of non-effectors vs $12/300 = 4\%$ of effectors).

→

According to this comment, we now revive this sentence but with a great tonedown. The original predictive statement together with a supplementary figure (listing 12 effectors containing the FxxxLxxxK sequence on a predicted α -helix near their C-terminus) is premature and could be misleading. To obtain exact numbers for calculating the enrichment, we searched the FxxxLxxxK sequence against the proteome of the strain *Legionella pneumophila* subsp. *pneumophila* str. Philadelphia 1 (Proteome ID: UP000000609), and noted that the previous number (more than 300 Legionella proteins) included redundant counting of the sequences that has the FxxxLxxxK sequence more than once. The correct number is 257.

It is now stated;

“257 out of a total of 2930 proteins of the *L. pneumophila* strain Philadelphia-1 (Proteome ID: UP000000609) contain the FxxxLxxxK sequence motif. Of these, 46 proteins, including SidH and VpdB, belong to the 280 known *Legionella* effectors (Inaba, Xu et al., 2019), indicating that this sequence is enriched in the effector proteins (16.4% of the effector proteins vs. 8.0% of the non-effector proteins).”